# Large Language Models Only Pass Primary School Exams in Indonesia: A Comprehensive Test on IndoMMLU

**Fajri Koto**[1]      **Nurul Aisyah**[2]      **Haonan Li**[1]      **Timothy Baldwin**[1,3]

[1]Department Natural Language Processing, MBZUAI
[2]Quantic School of Business and Technology
[3]The University of Melbourne

fajri.koto@mbzuai.ac.ae, nurulaisyah.inc@gmail.com, {haonan.li,timothy.baldwin}@mbzuai.ac.ae

## Abstract

Although large language models (LLMs) are often pre-trained on large-scale multilingual texts, their reasoning abilities and real-world knowledge are mainly evaluated based on English datasets. Assessing LLM capabilities beyond English is increasingly vital but hindered due to the lack of suitable datasets. In this work, we introduce IndoMMLU, the first multi-task language understanding benchmark for Indonesian culture and languages, which consists of questions from primary school to university entrance exams in Indonesia. By employing professional teachers, we obtain 14,981 questions across 64 tasks and education levels, with 46% of the questions focusing on assessing proficiency in the Indonesian language and knowledge of nine local languages and cultures in Indonesia. Our empirical evaluations show that GPT-3.5 only manages to pass the Indonesian primary school level, with limited knowledge of local Indonesian languages and culture. Other smaller models such as BLOOMZ and Falcon perform at even lower levels.[1]

## 1   Introduction

The evaluation of large language models (LLMs) has predominantly relied on English datasets to assess language proficiency (Wang et al., 2018; Baradaran et al., 2022), reasoning abilities (Zellers et al., 2019; Huang et al., 2019; Bisk et al., 2020; Talmor et al., 2019), and real-world knowledge (Hendrycks et al., 2021). LLMs such as GPT-3.5 (Ouyang et al., 2022), Falcon (Penedo et al., 2023), and BLOOMZ (Muennighoff et al., 2022), however, are pre-trained on large-scale multilingual data, and thus it is critical to evaluate what knowledge they capture and their reasoning abilities in languages beyond English.

School exams serve as a powerful means to assess the reasoning abilities and real-world knowl-

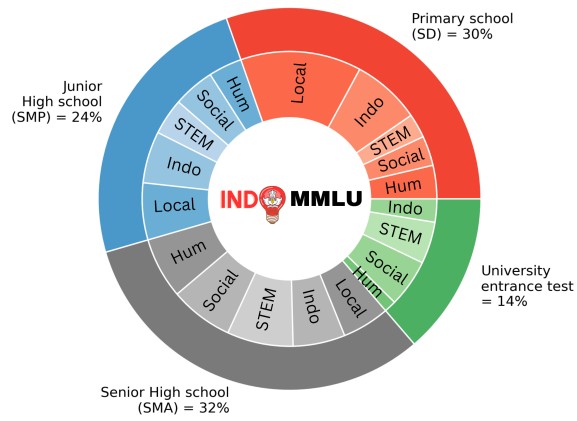

Figure 1: Distribution of subject areas and education levels in IndoMMLU. "Hum", "Social", "Indo", and "Local" refer to Humanities, Social Science, Indonesian Language, and Local Languages and Cultures, respectively.

edge of LLMs, given that these tests are meticulously designed by expert educators, drawing upon the principles of learning science. At various educational levels, school exams function as assessment tools, evaluating not only language proficiency but also higher-order cognitive skills such as comprehension, analytic abilities, and the application of real-world knowledge across diverse scenarios (Novak, 1988).

Hendrycks et al. (2021) proposed MMLU, a massive multitask language understanding benchmark in English that is compiled from different exams, covering topics including US history, computer science, and high school subjects. Recent progresses on LLMs such as LLaMA (Touvron et al., 2023) and GPT–4 (OpenAI, 2023) use MMLU as one of the evaluation datasets. In the GPT-4 technical report, automatic evaluation is further extended to encompass various standardized exams, including SAT, GRE, and bar exams.

While there has been a plethora of work on LLM evaluation for English (OpenAI, 2023; Katz et al.,

---

[1]Code and dataset can be found at https://github.com/fajri91/IndoMMLU

2023; Choi et al., 2023; Ryznar, 2023; Chalkidis, 2023), there has been comparatively little work in other languages (Li et al., 2023b; Sengupta et al., 2023). Recent work by OpenAI (2023) evaluated GPT-4 using a translated version of MMLU, and reported strong performance. While encouraging, using translations of English evaluation datasets has serious shortcomings, including translation noise, a complete lack of content that is sensitized to the local language/culture (esp. as most English evaluation datasets are highly US centric), and conversely, the existence of content that is irrelevant to the local language/culture (e.g. questions relating to US law or customs) and incongruent with the language-specific evaluation (Liu et al., 2023a).

In this paper, we ask professional teachers (of Indonesian nationality) to collect exam questions from various educational levels in Indonesian schools (i.e. primary school, junior high school, senior high school, and university). We categorize the collected questions into different subject areas, including: (1) STEM (Science, Technology, Engineering, and Mathematics); (2) Social Science; (3) Humanities; (4) Indonesian Language; and (5) Local Languages and Cultures. Figure 1 presents an overview of the distribution of the resulting dataset, IndoMMLU, across different subject areas and education levels. It is worth mentioning that 21% of the questions specifically focus on the Indonesian language, and 25% encompass nine distinct local languages and cultures that are specific to Indonesia.

Our contributions can be summarized as follows:

- We introduce the first Indonesian MMLU dataset, namely IndoMMLU, which comprises 64 tasks across different subject areas and education levels in Indonesia.

- Our dataset includes exam questions from school grades 1 to 12, as well as university entrance exams. This comprehensive coverage allows us to perform fine-grained assessment of the Indonesian language proficiency of existing LLMs.

- Approximately 25% of our data encompasses nine distinct local languages and cultures in Indonesia, namely Lampungic (ljp), Balinese (ban), Makassarese (mak), Banjarese (bjn), Madurese (mad), Sundanese (sun), Javanese (jav), Dayak Ngaju (nij), and Minangkabau.[2] These questions are not only in

under-represented languages but also incorporate specific cultural content, such as art, poetry, and daily life. For Lampungic (ljp) and Makassarese (mak) in particular, this is the very first NLP resource to be released.

- We evaluate various multilingual LLMs, including GPT-3.5 (Ouyang et al., 2022), XGLM (Lin et al., 2021), Falcon (Penedo et al., 2023), BLOOMZ (Muennighoff et al., 2022), mT0 (Muennighoff et al., 2022), LLaMA (Touvron et al., 2023), and Bactrian-X (Li et al., 2023a), across different model sizes. We find that only GPT-3.5 passes the highest primary school level exam, and no models demonstrate familiarity with local Indonesian languages and culture.

## 2 Related Work

**Evaluating Large Language Models** Various benchmarks have been released to evaluate English pre-trained LMs (Devlin et al., 2019; Conneau et al., 2020). Early benchmarks such as GLUE (Wang et al., 2018) and SuperGLUE (Wang et al., 2019) consist of various natural language understanding (NLU) tasks of different types with varying training data sizes. XGLUE (Liang et al., 2020), XTREME (Hu et al., 2020), and XTREME-R (Ruder et al., 2021) serve as multilingual benchmarks of more than 20 languages. For natural language generation (NLG), the GEM benchmark (Gehrmann et al., 2021) is a collection of machine translation, summarization, and generated descriptions in many languages.

As LLMs have become larger in size and improved over the standard benchmarks, there has been a shift in evaluation practice to focus on reasoning abilities (Zellers et al., 2019; Huang et al., 2019; Bisk et al., 2020; Talmor et al., 2019; Koto et al., 2022a), and real-world knowledge (Hendrycks et al., 2021). In GPT-4 (OpenAI, 2023), for instance, commonsense reasoning is evaluated using HellaSwag (Zellers et al., 2019) and WinoGrande (Sakaguchi et al., 2021), while real-world knowledge is evaluated based on school exams including MMLU (Hendrycks et al., 2021), ARC (Clark et al., 2018), and GSM-8K (Cobbe et al., 2021). Similarly, LLaMA (Touvron et al., 2023) was evaluated using school exam problems, in addition to closed-book question answering (Kwiatkowski et al., 2019; Joshi et al., 2017) and

---

[2]For Minangkabau culture, the Indonesian language is used in teaching and exams.

| Group | Question | Answer |
|---|---|---|
| Primary school | 99.6 | 65.5 |
| Junior high school | 188.3 | 105.9 |
| Senior high school | 167.7 | 130.3 |
| University Entrance Test | 204.9 | 186.2 |
| STEM | 133.7 | 102.4 |
| Social science | 136.2 | 131.9 |
| Humanities | 113.2 | 104.4 |
| Local languages and cultures | 88.4 | 68.0 |
| Indonesian language | 307.4 | 161.8 |

Table 1: Average question and answer length (in characters) for each education group and subject area.

the RACE reading comprehension benchmark (Lai et al., 2017).

**Indonesian Pre-trained Language Models and Benchmarks** Several monolingual pre-trained language models have been released for Indonesian, including IndoBERT (Koto et al., 2020b; Wilie et al., 2020), IndoBERTweet (Koto et al., 2021), and IndoBART (Cahyawijaya et al., 2021). These models have been evaluated on NLU (e.g. IndoLEM and IndoNLU) and NLG (e.g. IndoNLG) benchmarks. Component tasks include sentiment analysis (Koto and Rahmaningtyas, 2017; Purwarianti and Crisdayanti, 2019), emotion classification (Saputri et al., 2018), hate speech detection, summarization (Koto et al., 2020a, 2022b), and translation (Cahyawijaya et al., 2021; Koto and Koto, 2020).

In contemporaneous work, Cahyawijaya et al. (2023) evaluated several LLMs using existing Indonesian datasets. This collection includes the recent NusaX dataset (Winata et al., 2023), which is a parallel sentiment analysis dataset in 10 Indonesian local languages, created through human translation. The collection also includes several question-answering datasets, such as FactQA (Purwarianti et al., 2007), IDK-MRC (Putri and Oh, 2022), and TyDiQA (Clark et al., 2020), over news and Wikipedia documents. IndoMMLU is different in that it explicitly evaluates reasoning, language, and cultural abilities in a fine-grained manner from the perspective of education science.

## 3 IndoMMLU

IndoMMLU is a multiple-choice problem set in 64 subjects from different education levels, following the format of English MMLU (see Figure 2 and Figure 3). IndoMMLU, however, is based on the

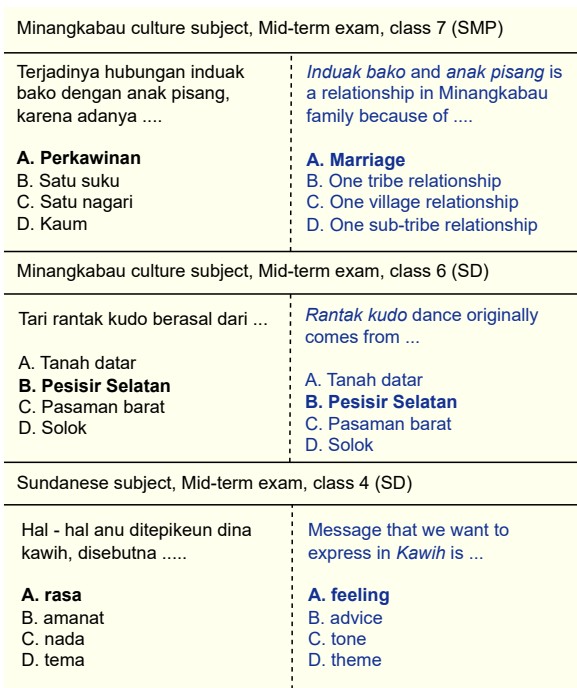

Figure 2: The first question focuses on the family relationship between *anak pisang* "children" and *induak bako* "aunt on the father's side". Both terms are commonly used in Minangkabau but not in the Indonesian language. The second and third questions pertain to traditional art. *Kawih* in the third question means a song set to a distinctive beat in Sundanese culture. **Left** is the original text and **right** is the English translation for illustrative purposes. The bold options are the correct answer keys.

Indonesian education curriculum, and has more fine-grained education levels than MMLU.

In Indonesia's curriculum, schools are categorized into three levels: (1) six years of primary school (*Sekolah Dasar* = "SD"), (2) three years of junior high school (*Sekolah Menengah Pertama* = "SMP"), and (3) three years of senior high school (*Sekolah Menengah Atas* = "SMA"). At primary school, pupils in all grades are taught the Indonesian language, civics, mathematics, art, sports, and religion. From grade 4 to 6 and in junior high school, pupils additionally learn a foreign language, a local language/culture, science, and social science.[3] In senior high school, pupils study more specialized natural science and social science subjects, including physics, chemistry, biology, geography, sociology, economy, and history. In IndoMMLU, we explicitly exclude mathematics because the questions typically consist primarily of symbols with lit-

---

[3] In a recent curriculum change, science and social science have been added from grade 3.

 **A. 17**
 B. 17 A
 C. 17 B
 D. 18
 E. 19 | Provisions regarding ministers are regulated in the 1945 Constitution of the Republic of Indonesia, namely in Chapter 5 concerning state ministries to be precise in the article...

 **A. 17**
 B. 17 A
 C. 17 B
 D. 18
 E. 19 |
| **Chemistry subject, University entrance test exam** | |
| Asam oksalat adalah asam berbasa dua. Sebanyak 10 mL larutan asam oksalat diencerkan dengan air sampai volumenya 100 mL. Larutan ini digunakan untuk menitrasi 20 mL larutan NaOH 0,2 M dengan indikator bromtimol biru. Bila titik akhir titrasi diperoleh saat volume asam oksalat mencapai 25 mL, maka konsentrasi larutan asam oksalat awal adalah...

Figure 3: Examples of civics and chemistry exam questions. **Left** is the original text and **right** is the English translation for illustrative purposes. The bolded options are the answer keys.

tle language content, and there are existing datasets for mathematical reasoning such as GSM-8K (Cobbe et al., 2021) and NumGLUE (Mishra et al., 2022).

The local language/culture subjects vary across provinces in Indonesia and depend on the local government policy. For example, in West Sumatra, Minangkabau culture is taught using the Indonesian language, while in West Java, pupils are exposed to the Sundanese language and culture. Figure 2 illustrates two exam questions for Minangkabau culture, and one exam question for Sundanese.

## 3.1 Data Construction

We asked seven professional teachers with at least a bachelor's degree in education to gather publicly-available school exam questions in Indonesia from web sources.[4] They were tasked with gathering problems for specific subject areas and educational levels, as well as metadata such as the source (i.e. URL of the source document), school level, class level, question, multiple-choice options, and the correct answer key. We instructed the teachers to

[4]The seven teachers were selected from 70 applicants.

| Group | Subjects |
|---|---|
| STEM | Chemistry (SMA, UE), Biology (SMA, UE), Physics (SMA, UE), Science (SD, SMP) |
| Social science | Geography (SMA, UE), Sociology (SMA, UE), Economy (SMA, UE), Civics education (SD, SMP, SMA), Social science (SD, SMP) |
| Humanities | History (SMA, UE), Art (SD, SMP, SMA), Sports (SD, SMP, SMA), Islam religion (SD, SMP, SMA), Christian religion (SD, SMP, SMA), Hindu religion (SD, SMP, SMA) |
| Local languages and cultures | Lampungic (SD, SMP, SMA), Balinese (SD, SMP, SMA), Makassarese (SD, SMP, SMA), Banjarese (SD, SMP, SMA), Madurese (SD, SMP, SMA), Minangkabau culture (SD, SMP), Dayak Ngaju (SD), Sundanese (SD, SMP, SMA), Javanese (SD, SMP, SMA) |
| Indonesian language | Indonesian language (SD, SMP, SMA, UE) |

Table 2: Subject areas in IndoMMLU. "SD", "SMP", "SMA", "UE" indicate that questions in the subject are are available in primary school, junior high school, senior high school, and university entrance exams, respectively.

only include exams that had accompanying answer keys, and to exclude problems that contained images. Additionally, we organized an 1-hour workshop to discuss the data collection procedure with all the teachers, addressing any questions or concerns they had. All teachers are paid competitively, higher than the Indonesian average monthly wage.[5]

## 3.2 Quality Control

To ensure the accuracy of the data entry process, we randomly checked questions collected by each teacher. We manually verified the questions, multiple-choice options, and the corresponding answer keys based on the given URL, and found that each teacher conducted the work accurately. We also additionally performed automatic filtering to remove repetitive questions, and remove questions that have no answer key.

## 3.3 Data Statistics

After data cleansing, we obtained a total of 14,981 questions, distributed over school levels and subjects as detailed in Figure 1; the details of each subject area are in Table 2 and the Appendix.

[5]The work for a single teacher was equivalent to a 5-day full-time job.

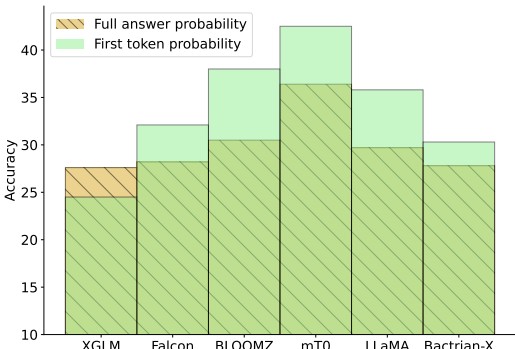

Figure 4: LLM performance (% accuracy) based on: (1) the probability of the full generated answer; and (2) the probability of the first token in the generated answer.

IndoMMLU consists of 30% primary school, 24% junior high school, 32% senior high school, and 14% university entrance exam questions. Table 1 shows the average question length for each education level and subject area. We can observe that primary school questions tend to be shorter and university entrance exam questions are longer. Indonesian language questions have the highest average length, while local languages and culture questions are around 88 characters on average.

## 4 Experiments

### 4.1 Set-Up

We evaluate 24 multilingual LLMs of different sizes in zero-shot and few-shot settings. This includes GPT-3.5 (Ouyang et al., 2022), XGLM (Lin et al., 2021), Falcon (Penedo et al., 2023), BLOOMZ (Muennighoff et al., 2022), mT0 (Muennighoff et al., 2022), LLaMA (Touvron et al., 2023) and Bactrian-X (Li et al., 2023a).[6] We add a simple prompt in the Indonesian language *Ini adalah soal [subject] untuk [level]. Pilihlah salah satu jawaban yang dianggap benar!* "This is a [subject] question for [level]. Please choose the correct answer!" prior to the question and multiple-choice options.

For closed-source models, we evaluate questions by comparing the first generated tokens (e.g., *A*, *B*, *C*) and the answer key using a regular expression.[7] For open-sourced models, we benchmark two strategies. Given a question and the corresponding multiple-choice options, we calculate: (1)

the probability of the full generated answer; and (2) the probability of the first token in the generated answer. For the first, we select the answer with the highest normalized log likelihood, and for the second, we simply select the key token (e.g., *C*) with the highest probability among all possible keys.

### 4.2 Results

Figure 4 presents the zero-shot accuracy when using: (1) the full answer probability; and (2) the probability of the first token in the generated answer. Among the open-sourced language models (LLMs) including XGLM (7.5B), Falcon (40B), BLOOMZ (7.1B), mT0$_{xxl}$ (13B), LLaMA (65B), and Bactrian-X (13B), we find that estimating the answer based on the probability of the first token in the generated answer generally performs best, with the notable exception of XGLM. Thus, we report results under this configuration in the remaining sections; the full results for both settings can be found in the Appendix.

**Results across all models** Table 3 shows the average accuracy for each subject area across the 24 models. To compute the scores, we disregard the education level of the questions, and average scores based on the subject (e.g. Biology), and finally combine the scores across all subject areas (e.g. STEM). The random performance varies between 20% to 27% due to the differing number of multiple-choice options (i.e. three to five).

Overall, we found that GPT-3.5 attains the highest overall accuracy, albeit low at 53.2%. GPT-3.5 is also notably the highest in each subject area, except in local languages and culture subjects. Among the open-source models, we observe that mT0$_{xxl}$ (13B) achieves an average accuracy of 42.5%. Falcon (40B) performs worse than mT0$_{xxl}$ (13B) and BLOOMZ (7B).

Performance based on model size varies, with smaller models such as BLOOMZ (7B) and mT0$_{xxl}$ being better than Falcon (40B) and LLaMA (65B). We suspect that this is due to the absence of the Indonesian language in Falcon and LLaMA's pre-training data. The poor performance of the 13B and 30B LLaMA models might imply that any "emergent abilities" of LLMs generally appear in the same or closely-related languages. This is further supported by Bactrian-X-LLaMA (13B), a LLaMA model fine-tuned on instruction datasets in 52 languages (including Indonesian), which obtain a +5% average increment, compared to LLaMA

---

[6]At the time this research was carried out, we did not have access to the GPT-4 API, and thus leave it to future work.

[7]In cases where there is no match, we assign a random answer.

| Model (#parameters) | STEM | Social Science | Humanities | Indonesian Language | Local languages and Cultures | Average |
|---|---|---|---|---|---|---|
| Random | 21.9 | 23.4 | 23.5 | 24.4 | 26.6 | 24.4 |
| GPT-3.5 (175B) | **54.3** | **62.5** | **64.0** | **62.2** | 39.3 | **53.2** |
| XGLM (564M) | 22.1 | 23.0 | 25.6 | 25.6 | 27.5 | 25.2 |
| XGLM (1.7B) | 20.9 | 23.0 | 24.6 | 24.8 | 26.6 | 24.4 |
| XGLM (2.9B) | 22.9 | 23.2 | 25.4 | 26.3 | 27.2 | 25.2 |
| XGLM (4.5B) | 21.8 | 23.1 | 25.6 | 25.8 | 27.1 | 25.0 |
| XGLM (7.5B) | 22.7 | 21.7 | 23.6 | 24.5 | 27.5 | 24.5 |
| Falcon (7B) | 22.1 | 22.9 | 25.5 | 25.7 | 27.5 | 25.1 |
| Falcon (40B) | 30.2 | 34.8 | 34.8 | 34.9 | 29.2 | 32.1 |
| BLOOMZ (560M) | 22.9 | 23.6 | 23.2 | 24.2 | 25.1 | 24.0 |
| BLOOMZ (1.1B) | 20.4 | 21.4 | 21.1 | 23.5 | 24.7 | 22.4 |
| BLOOMZ (1.7B) | 31.5 | 39.3 | 38.3 | 42.8 | 29.4 | 34.4 |
| BLOOMZ (3B) | 33.5 | 44.5 | 39.7 | 46.7 | 29.8 | 36.4 |
| BLOOMZ (7.1B) | 37.1 | 46.7 | 44.0 | 49.1 | 28.2 | 38.0 |
| mT0$_{small}$ (300M) | 21.8 | 21.4 | 25.7 | 25.1 | 27.6 | 24.9 |
| mT0$_{base}$ (580M) | 22.6 | 22.6 | 25.7 | 25.6 | 26.9 | 25.0 |
| mT0$_{large}$ (1.2B) | 22.0 | 23.4 | 25.1 | 27.3 | 27.6 | 25.2 |
| mT0$_{xl}$ (3.7B) | 31.4 | 42.9 | 41.0 | 47.8 | 35.7 | 38.2 |
| mT0$_{xxl}$ (13B) | 33.5 | 46.2 | 47.9 | 52.6 | **39.6** | 42.5 |
| LLaMA (7B) | 22.8 | 23.1 | 25.1 | 26.7 | 27.6 | 25.3 |
| LLaMA (13B) | 24.1 | 23.0 | 24.4 | 29.5 | 26.7 | 25.3 |
| LLaMA (30B) | 25.4 | 23.5 | 25.9 | 28.4 | 28.7 | 26.5 |
| LLaMA (65B) | 33.0 | 37.7 | 40.8 | 41.4 | 32.1 | 35.8 |
| Bactrian-X-LLaMA (7B) | 23.3 | 24.0 | 26.0 | 26.1 | 27.5 | 25.7 |
| Bactrian-X-LLaMA (13B) | 28.3 | 29.9 | 32.8 | 35.2 | 29.2 | 30.3 |

Table 3: Zero-shot performance (% accuracy) of LLMs, combined across education levels. "Average" means the average across all subject areas in IndoMMLU.

(13B).

**Results across education levels** As illustrated in Figure 1, IndoMMLU includes detailed education level metadata, which enables us to gain a deeper understanding of the capabilities of LLMs in terms of human education levels. In the Indonesian context, the minimum passing score for exams varies across subjects and typically ranges between 65 and 70.[8] By setting the passing score at 65, we assess GPT-3.5 over real-world knowledge capabilities, as shown in Table 4. Green indicates that the model has successfully passed the subject, while red indicates it has failed. This reveals that GPT-3.5 generally performs well on primary school exams for general subjects, but exhibits a lack of understanding of local languages and culture. In subjects that require less analytical thinking, such as civics and religion, GPT-3.5 tends to achieve higher scores in high school exams.

**Indonesian language proficiency of LLMs** As discussed in Section 3, IndoMMLU specifically includes Indonesian language exams for all grades

[8]This refers to Curriculum 2013 in Indonesia.

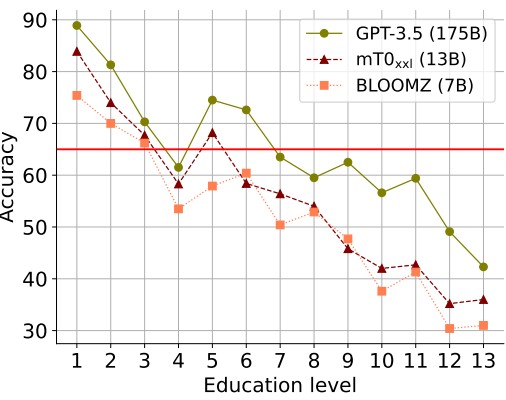

Figure 5: Fine-grained accuracy (%) of GPT-3.5, mT0$_{xxl}$, and BLOOMZ in the Indonesian language subject area. The horizontal line depicts the passing score of 65, and the education level of 13 refers to the university entrance exam.

and education levels, allowing us to assess the Indonesian language proficiency of LLMs. Figure 5 illustrates that GPT-3.5 achieves its highest accuracy in grade 1, approaching 90%. However, as the education level increases, the model's performance gradually declines. For grades 3 and above, the

| Subject | SD | SMP | SMA | UE |
|---|---|---|---|---|
| Science | 76.3 | 67.8 | 52.8 | 43.7 |
| Social science | 84.6 | 73.1 | 63.5 | 48.2 |
| Indonesian language | 74.7 | 61.8 | 55.1 | 42.3 |
| Civics | 64.6 | 65.2 | 65.4 | – |
| Sports | 66.7 | 44.7 | 62.0 | – |
| Art | 73.9 | 71.2 | 58.7 | – |
| Islam religion | 78.6 | 59.9 | 67.7 | – |
| Christian religion | 83.7 | 77.6 | 62.0 | – |
| Hindu religion | 66.7 | 62.0 | 55.1 | – |
| Sundanese | 50.0 | 45.1 | 37.9 | – |
| Javenese | 46.1 | 36.1 | 36.1 | – |
| Balinese | 32.2 | 38.5 | 36.1 | – |
| Makassarese | 33.7 | 48.8 | 38.3 | – |
| Banjarese | 50.0 | 44.4 | 28.6 | – |
| Lampungic | 40.0 | 30.0 | 33.3 | – |
| Madurese | 41.0 | 28.3 | 35.0 | – |
| Minangkabau culture | 38.0 | 52.2 | – | – |
| Dayak Ngaju | 31.1 | – | – | – |

Table 4: GPT-3.5 performance (% accuracy) across different education levels. "SD", "SMP", "SMA", "UE" indicate primary school, junior high school, senior high school, and university entrance tests, respectively. Red indicates that the score is below the minimum passing threshold of 65, while green signifies a score at or above this minimum.

scores fall below 75, and for classes 7 and above, GPT-3.5 fails to pass the exams. We observe that this trend is similar for mT0$_{xxl}$ and BLOOMZ, which only pass grades 1, 2, and 3. This fine-grained evaluation provides a valuable benchmark for LLM proficiency in Indonesian.

**LLM performance on local languages and cultures** It is interesting to observe in Table 3 that despite having only 13B parameters, mT0$_{xxl}$ achieves the highest accuracy on local languages and cultures. On the other hand, GPT-3.5 with 175B parameters achieves competitive accuracy, just 0.3 absolute points lower than mT0$_{xxl}$. To further investigate this, Figure 6 displays the accuracy scores of each local language and culture subject, revealing that both mT0$_{xxl}$ and GPT-3.5 excel in different subject areas. mT0$_{xxl}$ shows greater familiarity with Javanese and Sundanese, with a disparity of +10 for both subjects compared to GPT-3.5. GPT-3.5 performs better in Dayak Ngaju, Banjarese, and Minangkabau culture.

### 4.3 Analysis

**Few shot performance** Providing several questions and the answer key in prompts has been re-

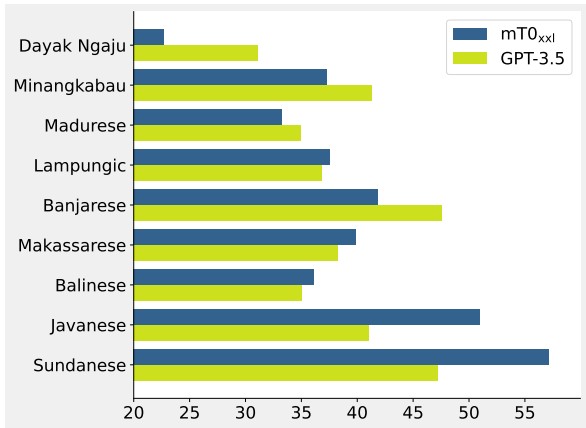

Figure 6: Zero-shot performance of mT0$_{xxl}$ and GPT-3.5 in local languages and cultures subjects.

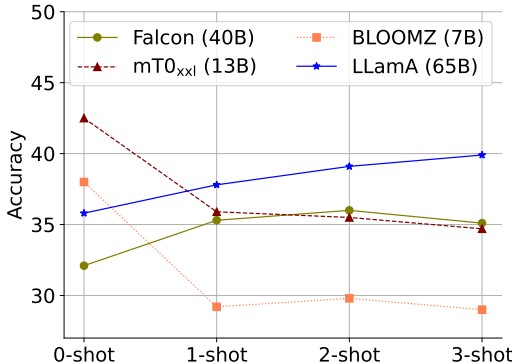

Figure 7: Few-shot performance (% accuracy) of mT0$_{xxl}$, BLOOMZ, Falcon, and LLaMA, averaged across all subject areas.

ported to improve model performance (Karimi Mahabadi et al., 2022; Hendrycks et al., 2021). We run similar experiments with our top-4 best open-source models and observe mixed outcomes in Figure 7.[9] Few-shot inference does not yield improvements in instruction-tuned models like mT0 and BLOOMZ, as evidenced by a decrease in accuracy. In contrast, the pure LLMs Falcon and LLaMA show better performance with few-shot inference compared to zero-shot. These findings align with those of Liu et al. (2023b); Li et al. (2023b), where few-shot prompts may lead to unnatural inferences for instruction-tuned models.

**Model confidence** Given the top three models in Table 3, we assess whether their confidence predictions (i.e. the predicted likelihood of the predicted answer being correct) corresponds to the actual accuracy across 64 tasks. This uncertainty calibration gives us hints about the model's reliability

---

[9]Refer to the Appendix for details of the prompts.

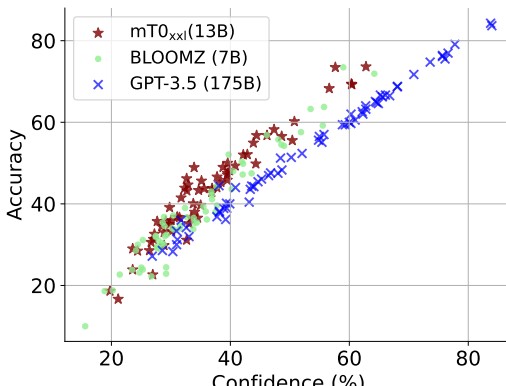

Figure 8: Zero-shot calibration of mT0$_{xxl}$, BLOOMZ, and GPT-3.5 across 64 tasks. The average standard deviations of the confidence scores across all data points are 36.5, 26.4, and 43.9, respectively

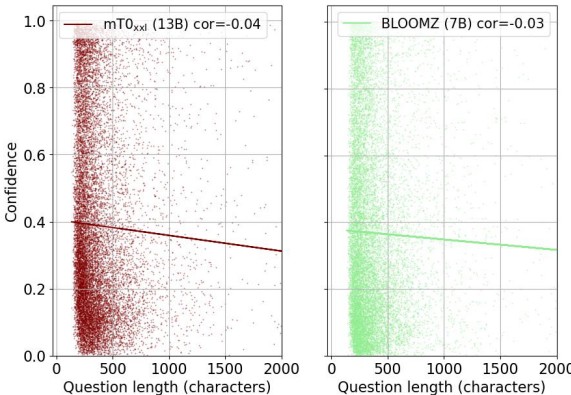

Figure 9: Correlation between question difficulty and question length.

| Model | W/ negation | W/o negation |
|---|---|---|
| *Indonesian language* | | |
| GPT-3.5 (175B) | 58.0 | **62.7** |
| mT0$_{xxl}$ (13B) | 47.9 | **53.1** |
| BLOOMZ (7B) | 39.3 | **50.1** |
| *Social science* | | |
| GPT-3.5 (175B) | **66.2** | 63.0 |
| mT0$_{xxl}$ (13B) | **48.2** | 47.1 |
| BLOOMZ (7B) | 43.3 | **48.2** |

Table 5: Accuracy (%) for questions with and without negation in the Indonesian language and social science subject areas.

and how to use them appropriately in real-world settings. For mT0 and BLOOMZ, the confidence score is determined through softmax normalization over probabilities of the multiple-choice options. For GPT-3.5, we adopt the approach described by Si et al. (2022); Wang et al. (2022), using a high-temperature value (0.7) during decoding. For each question, we generate $n$ different outputs and measure self-consistency. The probability of a multiple-choice option is calculated based on the output frequency. In this experiment, we use $n = 7$, and choose the most frequently-occurring answer as the final prediction.

We average the confidence scores across the 64 tasks, and display the calibration of mT0, BLOOMZ, and GPT-3.5 in Figure 8. We observe that all three models are well-calibrated, with correlation scores of $r > 0.85$.

Additionally, we examine the relationship between confidence scores and question length, as depicted in Figure 9. We found a very weak correlation for both mT0 and BLOOMZ. It is worth noting that the confidence score can also be interpreted as a measure of question difficulty, based on which question length appears to have no bearing on difficulty.

**Impact of negation** In Indonesian school exam questions, the use of negation is common to enhance question difficulty and assess students' reasoning abilities. Similarly, in the field of NLP, negation is known to increase the difficulty of NLP tasks (Truong et al., 2022). To investigate the impact of negation, we employ a simple string-matching strategy to identify questions that contain negations

within each subject area.[10] We then break down the accuracy for the top three models (GPT-3.5, mT0, and BLOOMZ) based on the presence or absence of negation. Among the subject areas, Indonesian language and social science are the most prevalent in employing negation, accounting for approximately 10% in each group. Through manual observation of 100 random samples, we verified that 85% of these questions indeed contained negation.

Table 5 shows the effects of negation on IndoMMLU accuracy. For the Indonesian language subject area, negated questions prove to be more challenging, with a decrease in accuracy ranging from $-4$ to $-10$. In social science, mT0 and BLOOMZ are similarly more accurate over questions without negation. Compared to mT0, however, BLOOMZ is less robust to negation, as indicated by the $-5$ accuracy drop.

---

[10] To identify negation, we use strings *kecuali* "except", *yang bukan* "which is not", and *yang tidak* "which is not".

## 5 Discussion

If LLMs are to be deployed in diverse contexts, it is critical to have more work on evaluation for different languages and cultures. In Table 1 we observed that the models struggle to answer questions that pertain to local languages and cultures across all levels of education in Indonesia. Minangkabau culture in particular is taught and assessed in the Indonesian language, and yet the limited performance in answering questions relating to it underscores a lack of cultural knowledge, despite reasonable results for the Indonesian language.

We also argue that education science should play a more central role in the future evaluation of LLMs. Current NLP work has mostly focused on developing larger models with different techniques and architectures, and evaluation has primarily been in terms of specific NLP tasks. Education science has decades of experience in designing assessments to evaluate student progress through painstakingly-designed comprehensive tests, which the NLP community should better engage with. With IndoMMLU, we have shown that exam questions across fine-grained educational levels offer a more profound comprehension of model proficiency in the Indonesian language, while also revealing potential areas for improvement.

## 6 Conclusion

In this paper, we presented IndoMMLU, a multi-task language understanding benchmark for real-world evaluation of knowledge in the Indonesian context. By leveraging education level metadata, we found that current LLMs like GPT-3.5 are only able to pass primary school exams in Indonesia, while smaller models struggle across nearly in all education levels. Notably, none of the 24 evaluated models perform well in the domain of local languages and cultures, highlighting the need for further research in this direction.

## Limitations

Despite being the largest question-answering dataset in the Indonesian context, IndoMMLU still has some limitations, in that it lacks: (1) multimodal questions; (2) arithmetic reasoning tasks; and (3) essay-style questions. First, IndoMMLU is comprised solely of text-based questions, and questions with tables and figures are discarded to simplify data collection. We specifically exclude math questions as they are already well covered by existing English math reasoning benchmarks. We suggest that essay questions enable a deeper assessment of comprehension and critical thinking, but that methods for evaluating essay quality across education levels in languages other than English are severely lacking.

## Ethical Considerations

The IndoMMLU dataset used in our study is collected from publicly-available web resources. In compliance with the Indonesian Copyright Law number 28 year 2014, specifically article 44, the use, retrieval, reproduction, and/or modification of works and/or related rights products, in whole or in substantial part, is not considered a copyright infringement if the source is fully cited or mentioned for educational and research purposes.[11]

Regarding our experimental results, it is important to note that they do not provide a definitive answer as to the relative abilities of LLMs, and we caution readers against overinterpreting the findings. While we conclude that GPT-3.5 demonstrates proficiency in passing primary school exams in Indonesia based on IndoMMLU, it is essential to consider potential contamination in GPT-3.5's pre-training data, which could impact the results. Furthermore, it is worth noting that real-world student assessments encompass not only multiple-choice questions but also practical exams, laboratory work, and essay writing.

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

## A Data Statistics

Table 6, Table 7, and Table 8 provide detailed statistics of the question distribution in IndoMMLU.

| Subjects | SD | SMP | SMA | UE | Total |
|---|---|---|---|---|---|
| Science | 488 | 680 | – | – | 1168 |
| Physics | – | – | 297 | – | 297 |
| Chemistry | – | – | 287 | 398 | 685 |
| Biology | – | – | 457 | 388 | 845 |
| Social science | 300 | 299 | – | – | 599 |
| Geography | – | – | 196 | 294 | 490 |
| Sociology | – | – | 295 | 201 | 496 |
| Economics | – | – | 296 | 192 | 488 |
| History | – | – | 300 | 198 | 498 |
| Civics | 99 | 300 | 300 | – | 699 |
| Indonesian language | 1125 | 850 | 857 | 381 | 3213 |
| Balinese | 200 | 123 | 148 | – | 471 |
| Makassarese | 98 | 41 | 47 | – | 186 |
| Banjarese | 120 | 10 | 14 | – | 144 |
| Lampungic | 93 | 30 | 24 | – | 147 |
| Madurese | 100 | 93 | 102 | – | 295 |
| Sundanese | 718 | 294 | 145 | – | 1157 |
| Javanese | 396 | 298 | 298 | – | 992 |
| Dayak Ngaju | 109 | – | – | – | 109 |
| Minangkabau culture | 153 | 46 | – | – | 199 |
| Art | 200 | 200 | 201 | – | 601 |
| Sports | 49 | 49 | 50 | – | 148 |
| Islam religion | 201 | 202 | 300 | – | 703 |
| Christian religion | 50 | 49 | 102 | – | 201 |
| Hindu religion | 49 | 52 | 49 | – | 150 |
| Total | 4548 | 3616 | 4765 | 2052 | 14981 |

Table 6: Total number of questions for each subject area and education level. "SD", "SMP", "SMA", "UE" indicate primary school, junior high school, senior high school, and university entrance tests, respectively.

| Class | #questions |
|---|---|
| 1 | 200 |
| 2 | 150 |
| 3 | 195 |
| 4 | 187 |
| 5 | 196 |
| 6 | 197 |
| 7 | 282 |
| 8 | 291 |
| 9 | 277 |
| 10 | 295 |
| 11 | 288 |
| 12 | 274 |
| 12+ | 381 |
| Total | 3213 |

Table 7: Total number of questions in the Indonesian language subject, including those designated for university entrance tests (12+).

| Category | #question |
|---|---|
| STEM | 2995 |
| Social science | 2772 |
| Humanities | 2301 |
| Indonesian language | 3213 |
| Local languages and cultures | 3700 |
| Total | 14981 |

Table 8: Total number of questions based on subject areas.

## B Few-shot Prompt

| Ini adalah beberapa contoh soal [SUBJECT]. | These are several examples of [SUBJECT] question. |
|---|---|
| [Example-1] Jawaban: [Answer-1] | [Example-1] Answer: [Answer-1] |
| [Example-2] Jawaban: [Answer-2] | [Example-2] Answer: [Answer-2] |
| [Example-3] Jawaban: [Answer-3] | [Example-3] Answer: [Answer-3] |
| [QUESTION] Jawaban: | [QUESTION] Answer: |

Figure 10: Illustration of our few-shot prompt template. The English translation on the right is solely for illustrative purposes. In our experiments, we used up to three examples within the prompt. The placeholders [SUBJECT], Example-i, Answer-i, and QUESTION correspond to the subject, the $i$-th question example, the answer key for the $i$-th question example, and the main question, respectively.

## C Zero-shot Performance Based on the Probability of the Full Generated Answer

| Model (#parameters) | STEM | Social Science | Humanities | Indonesian Language | Local languages and Cultures | Average |
|---|---|---|---|---|---|---|
| Random | 21.9 | 23.4 | 23.5 | 24.4 | 26.6 | 24.4 |
| XGLM (564M) | 24.2 | 25.9 | 27.2 | 29.0 | 27.8 | 26.8 |
| XGLM (1.7B) | 23.7 | 25.4 | 27.1 | 28.4 | 28.9 | 26.9 |
| XGLM (2.9B) | 23.6 | 25.4 | 28.3 | 28.8 | 28.8 | 26.9 |
| XGLM (4.5B) | 23.9 | 25.5 | 29.4 | 27.9 | 28.1 | 27.2 |
| XGLM (7.5B) | 23.5 | 26.0 | 29.4 | 28.6 | 28.9 | 27.6 |
| Falcon (7B) | 22.2 | 25.8 | 28.4 | 30.1 | 27.9 | 26.8 |
| Falcon (40B) | 25.8 | 28.4 | 29.5 | 32.9 | 27.7 | 28.2 |
| BLOOMZ (560M) | 23.0 | 24.4 | 23.7 | 27.2 | 26.4 | 24.9 |
| BLOOMZ (1.1B) | 22.9 | 25.8 | 26.6 | 28.3 | 27.4 | 26.2 |
| BLOOMZ (1.7B) | 23.7 | 29.8 | 29.7 | 32.8 | 28.1 | 28.3 |
| BLOOMZ (3B) | 27.6 | 32.5 | 32.6 | 35.0 | 27.4 | 30.0 |
| BLOOMZ (7.1B) | 26.8 | 32.9 | 33.5 | 36.5 | 28.1 | 30.5 |
| mT0$_{small}$ (300M) | 24.0 | 26.1 | 27.0 | 29.8 | 30.8 | 27.8 |
| mT0$_{base}$ (580M) | 23.9 | 25.5 | 27.6 | 30.1 | 30.5 | 27.7 |
| mT0$_{large}$ (1.2B) | 25.1 | 27.5 | 27.9 | 33.6 | 29.6 | 28.2 |
| mT0$_{xl}$ (3.7B) | 28.5 | 36.1 | 35.3 | 40.7 | 34.3 | 34.2 |
| mT0$_{xxl}$ (13B) | 30.1 | 38.1 | 40.9 | 43.2 | 34.5 | 36.4 |
| LLamA (7B) | 23.7 | 25.6 | 28.0 | 29.0 | 28.3 | 27.0 |
| LLamA (13B) | 24.0 | 25.4 | 27.7 | 29.4 | 29.6 | 27.4 |
| LLamA (30B) | 24.3 | 26.4 | 29.5 | 29.8 | 28.5 | 27.7 |
| LLamA (65B) | 26.7 | 29.3 | 32.4 | 32.9 | 29.0 | 29.7 |
| Bactrian-X-LLamA (7B) | 23.8 | 25.4 | 28.7 | 29.8 | 28.0 | 27.0 |
| Bactrian-X-LLamA (13B) | 25.6 | 27.4 | 29.2 | 30.7 | 27.9 | 27.8 |

Table 9: Zero-shot performance (% accuracy) of large language models based on **the probability of the full generated answer**, aggregated across education levels. "Average" means the average across all subject areas in `IndoMMLU`.

## D Model Artifacts

| Models (#parameters) | Source |
|---|---|
| XGLM (564M) | facebook/xglm-564M |
| XGLM (1.7B) | facebook/xglm-1.7B |
| XGLM (2.9B) | facebook/xglm-2.9B |
| XGLM (4.5B) | facebook/xglm-4.5B |
| XGLM (7.5B) | facebook/xglm-7.5B |
| Falcon (7B) | tiiuae/falcon-7b |
| Falcon (40B) | tiiuae/falcon-40b |
| BLOOMZ (560M) | bigscience/bloomz-560m |
| BLOOMZ (1.1B) | bigscience/bloomz-1b1 |
| BLOOMZ (1.7B) | bigscience/bloomz-1b7 |
| BLOOMZ (3B) | bigscience/bloomz-3b |
| BLOOMZ (7.1B) | bigscience/bloomz-7b1 |
| mT0$_{small}$ (300M) | bigscience/mt0-small |
| mT0$_{base}$ (580M) | bigscience/mt0-base |
| mT0$_{large}$ (1.2B) | bigscience/mt0-large |
| mT0$_{xl}$ (3.7B) | bigscience/mt0-xl |
| mT0$_{xxl}$ (13B) | bigscience/mt0-xxl |
| LLamA (7B) | decapoda-research/llama-7b-hf |
| LLamA (13B) | decapoda-research/llama-13b-hf |
| LLamA (30B) | decapoda-research/llama-30b-hf |
| LLamA (65B) | huggyllama/llama-65b |
| Bactrian-X-LLamA (7B) | MBZUAI/bactrian-x-llama-7b-lora |
| Bactrian-X-LLamA (13B) | MBZUAI/bactrian-x-llama-13b-lora |

Table 10: With the exception of GPT–3.5 (Ouyang et al., 2022), all the models used in this study were sourced from Huggingface (Wolf et al., 2020).

## E  Full Results in Each Subject and Education Level in GPT-3.5, mT0, and BLOOMZ

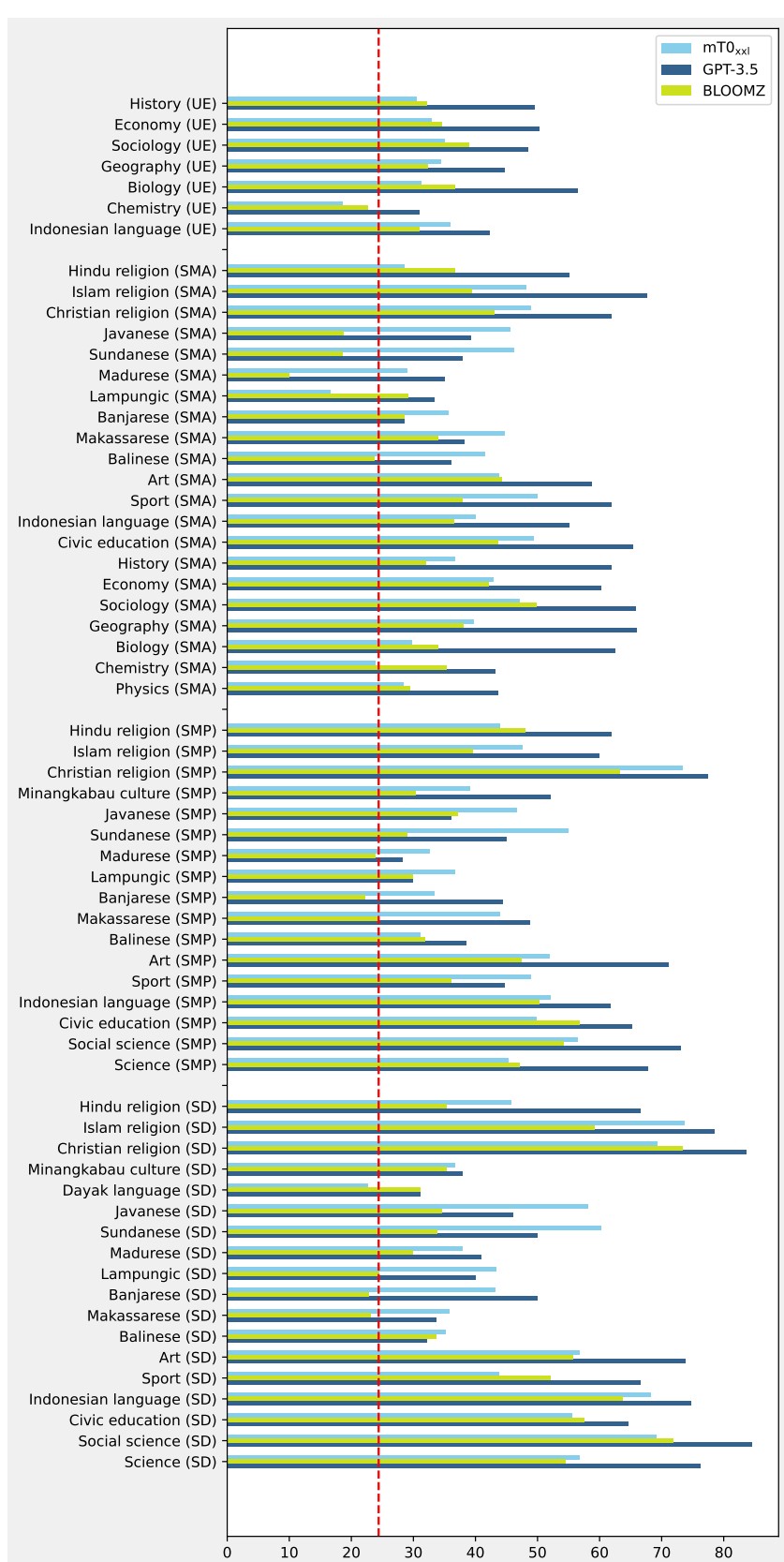

Figure 11: Performance (% accuracy) breakdown across the 64 tasks. "SD", "SMP", "SMA", "UE" indicate primary school, junior high school, senior high school, and university entrance tests, respectively. The red vertical line denotes random performance.