# OpenReview forum: "Large Language Models Only Pass Primary School Exams in Indonesia: A Comprehensive Test on IndoMMLU"
_EMNLP/2023/Conference — EMNLP 2023 Main_

### Official Review · Reviewer_49YF · 2023-08-04

**Soundness:** 4

**Excitement:**

4: Strong: This paper deepens the understanding of some phenomenon or lowers the barriers to an existing research direction.

**Paper Topic And Main Contributions:**

This paper introduced a massive multitask language understanding benchmark for Large Language Models (LLMs) in Indonesian languages. The proposed benchmark included exam questions for K-12 school students covering multiple subjects. The primary contribution of this study lay in the creation of a benchmark for minor languages. The proposed benchmark was thoroughly examined with 24 multilingual LLMs, and the results were compared across the models and education levels. Additionally, this study analyzed the few-shot performance of LLMs on this benchmark. The findings aligned with previous studies, indicating that state-of-the-art LLMs perform poorly in Indonesian languages.

**Reasons To Accept:**

This study effectively developed a benchmark for LLMs in Indonesian languages by using K-12 exams. Particularly, this benchmark demonstrated that LLMs encountered difficulties in answering questions related to local languages and cultures, resulting in inconsistent performances across the board.

**Reasons To Reject:**

This study could have verified the proposed benchmark's applicability across languages, such as English and Indonesian languages. This expansion would lead to a more comprehensive evaluation.

**Reproducibility:**

3: Could reproduce the results with some difficulty. The settings of parameters are underspecified or subjectively determined; the training/evaluation data are not widely available.

**Reviewer Confidence:**

3: Pretty sure, but there's a chance I missed something. Although I have a good feel for this area in general, I did not carefully check the paper's details, e.g., the math, experimental design, or novelty.

---

> ### Author Rebuttal · Authors · 2023-08-29
>
> Thank you for your positive feedback.
>
> We completely agree that applicability analysis across languages is interesting. One way to do it is by translating IndoMMLU to English via translation. However, 25% of IndoMMLU uses 10 Indonesian local languages. Finding good human/machine translators is extremely challenging for these local languages.

---

### Official Review · Reviewer_AUi2 · 2023-08-05

**Soundness:** 4

**Excitement:**

3: Ambivalent: It has merits (e.g., it reports state-of-the-art results, the idea is nice), but there are key weaknesses (e.g., it describes incremental work), and it can significantly benefit from another round of revision. However, I won't object to accepting it if my co-reviewers champion it.

**Paper Topic And Main Contributions:**

This study proposes a benchmark to investigate the performance of large-scale language models using school exams in Indonesian.
It is an excellent attempt to investigate the performance of large-scale language models in languages other than English.

**Questions For The Authors:**

Some of the questions include questions about Indonesian culture and geography. I think it is of important interest to the reader whether the inclusion of questions about Indonesian culture and geography is the cause of the poor performance of the LLM or whether the problem is simply that there are too few Indonesian language resources. I believe that asking questions about Indonesian culture and geography in English would yield suggestive results in this regard. What would be the contribution of this study for this point?

**Reasons To Accept:**

The attempt to investigate the performance of large-scale language models in Indonesian language. It can alleviate the English-centricity of natural language processing.

**Reasons To Reject:**

There is no evaluation using GPT-4, which seems to perform the best, even though the main result is the claim "Large Language Models Only Pass Primary School Exams in Indonesia," which suggests performance limitations.
"We do not have access to the GPT-4 API, thus leave it for future work. However, even if we do not have access to the API, it should be possible to ask the GPT-4 about 200 questions and collect the answers via a form. If the main claim is the limitation of performance, an evaluation using the method that is considered to have the best performance should be added, even if it is only a small-scale experiment.

**Reproducibility:**

3: Could reproduce the results with some difficulty. The settings of parameters are underspecified or subjectively determined; the training/evaluation data are not widely available.

**Reviewer Confidence:**

3: Pretty sure, but there's a chance I missed something. Although I have a good feel for this area in general, I did not carefully check the paper's details, e.g., the math, experimental design, or novelty.

---

> ### Author Rebuttal · Authors · 2023-08-29
>
> Thank you for your positive and constructive feedback.
>
> > There is no evaluation using GPT-4
>
> A: We agree that it would be interesting to add GPT-4 evaluation. However, we do not have access to GPT-4 prior to submission. Furthermore, we argue that evaluating a closed-source model such as GPT-4 provides less benefit to the research community, in part because of the difficulty in reproducing the results because of behind-the-scenes model updates. Nonetheless, we can certainly add GPT-4 to the camera-ready paper, as we now have access to the model.
>
> > I believe that asking questions about Indonesian culture and geography in English would yield suggestive results in this regard. What would be the contribution of this study for this point?
>
> A: We agree that it will be a valuable discussion. However, the questions for Indonesian culture subjects are in the local languages, which are extremely challenging to translate into English because they are extremely low resource. Moreover, our focus in this paper is to evaluate knowledge embedded in Indonesian models.

---

### Official Review · Reviewer_Qs8E · 2023-08-10

**Soundness:** 4

**Excitement:**

3: Ambivalent: It has merits (e.g., it reports state-of-the-art results, the idea is nice), but there are key weaknesses (e.g., it describes incremental work), and it can significantly benefit from another round of revision. However, I won't object to accepting it if my co-reviewers champion it.

**Paper Topic And Main Contributions:**

The paper introduces IndoMMLU, a multi-task language understanding for Indonesian. This dataset includes 14,906 exam questions from grades 1 to 12 and the university entrance exam. The questions within IndoMMLU encompass various subjects taught in Indonesian educational institutions, including STEM, Social Science, Humanities, local languages and cultures, and the Indonesian language. The construction process of IndoMMLU is meticulously documented, ensuring transparency and reliability. Subsequently, this paper also comprehensively analyzes the performance of 24 large language models (LLMs) on IndoMMLU.

**Questions For The Authors:**

Question A: (Section Discussion) Why should education science play a more central role? The experiments and results of this paper do not provide persuasive proof for this argument. To strengthen the standpoint of evaluating models on a learning curve, the author(s) should explain why the knowledge of LLMs in grade 12 should be disregarded if they fail in grade 4. For example, if an LLM fails the test about basic fractional operations, we can reasonably disregard the performance of this LLM on calculus. However, this approach does not appear to be reasonably applied to subjects such as Social Science or Humanities  where the progression of knowledge is not consistently cumulative.

This direction may be linked to the idea in procedural understanding of scientific facts in paper Tracking State Changes in Procedural Text: a Challenge Dataset and Models for Process Paragraph Comprehension (Dalvi et al., NAACL 2018). Besides, the discussion of Creating Dependency Between Questions in paper Benchmarking Machine Reading Comprehension: A Psychological (Sugawara et al., EACL 2021) may also offer some insights.

**Reasons To Accept:**

1.	IndoMMLU is a valuable resource for evaluating the reasoning abilities in Indonesian languages and real-world knowledge about Indonesia.

2.	The comprehensive analysis in Section 4 uncovers many significant insights about different LLMs' abilities and real-world knowledge.

**Reasons To Reject:**

1.	The benefits of evaluating LLMs on a learning curve (from 1 to 12) still need to be further explored in the current version of the paper.

Refer to the Question A.

**Reproducibility:**

3: Could reproduce the results with some difficulty. The settings of parameters are underspecified or subjectively determined; the training/evaluation data are not widely available.

**Reviewer Confidence:**

3: Pretty sure, but there's a chance I missed something. Although I have a good feel for this area in general, I did not carefully check the paper's details, e.g., the math, experimental design, or novelty.

---

> ### Author Rebuttal · Authors · 2023-08-29
>
> Thank you for the constructive feedback.
>
> > The benefits of evaluating LLMs on a learning curve (from 1 to 12) still need to be further explored
>
> A: Our focus is not on the “learning curve” and we do not make any strong claims about the monotonicity of results across grade levels (as in, a score of x at a lower grade level must mean a score of <= x for a higher grade level), but on (1) understanding how well a given large language model embeds knowledge across different subjects and pegged against different education levels, to get more motivated and fine-grained insights into model performance; and (2) creating a high-quality dataset that is multi-task (with good coverage across different tasks), crafted by education experts, and covers the Indonesian language in addition to 10 local languages. The Lampungic and Makassarese questions in IndoMMLU are the very first NLP resources to be released for these languages.
>
> > Why should education science play a more central role?
>
> A: As stated in L.479-483, we draw on education science to evaluate knowledge embedded in large language models. School assessments (exam questions) are not designed in an ad hoc way by crowd workers or drawn from arbitrary web sources, but are expertly based on a curriculum in education science, to evaluate the degree of mastery of the curriculum content by a student of a given grade level. In terms of knowledge acquisition by large language models, this provides fine-grained insights into model limitations that can be used to drive further improvements.
>
> > Paper references from NAACL 2018 and EACL 2021.
>
> A: These papers introduce reading comprehension tasks, which are not directly related to the primary focus of our work in evaluating the knowledge of LLMs in an Indonesian context, but we can certainly add them to the final version of the paper.

---

### Meta-Review · Area_Chair_WAQF · 2023-09-19

**Recommendation:** 4

**Metareview:**

This work presents a benchmark using K-12 exams for LLMs in Indonesian languages. It demonstrated that LLMs encountered difficulties in answering questions related to local languages and cultures. The reviewers agree that the dataset, i.e. IndoMMLU, is a valuable resource for evaluating the reasoning abilities in Indonesian languages and real-world knowledge about Indonesia. The comprehensive analysis showed interesting insights about different LLMs' abilities. Another good aspect is that this research is on promoting AI democratization, i.e. drawing attention to the LLMs' limitation on low-resource languages.

---

### Decision · Program_Chairs · 2023-10-07

**Decision:**

Accept-Main

**Comment:**

This work presents a benchmark using K-12 exams for LLMs in Indonesian languages. It demonstrated that LLMs encountered difficulties in answering questions related to local languages and cultures. The reviewers agree that the dataset, i.e. IndoMMLU, is a valuable resource for evaluating the reasoning abilities in Indonesian languages and real-world knowledge about Indonesia. The comprehensive analysis showed interesting insights about different LLMs' abilities. Another good aspect is that this research is on promoting AI democratization, i.e. drawing attention to the LLMs' limitation on low-resource languages.